

# Spatial variability of microzooplankton grazing on phytoplankton in coastal southern Florida, USA

Nicole Millette[1], Christopher Kelble[2], Ian Smith[2,3], Kelly Montenero[2,3] and Elizabeth Harvey[4]

[1] William & Mary, Virginia Institute of Marine Science, Gloucester Point, VA, United States
[2] Atlantic Oceanographic and Meteorological Laboratory, National Oceanic and Atmospheric Administration, Miami, FL, United States
[3] University of Miami, Cooperative Institute for Marine and Atmospheric Studies, Miami, FL, United States
[4] Department of Biological Sciences, University of New Hampshire, Durham, NH, United States

## ABSTRACT

Microzooplankton are considered the primary consumers of phytoplankton in marine environments. Microzooplankton grazing rates on phytoplankton have been studied across the globe, but there are still large regions of the ocean that are understudied, such as sub-tropical coastal oceans. One of these regions is the coastal area around south Florida, USA. We measured microzooplankton grazing rates in two distinct environments around south Florida; the oligotrophic Florida Keys and the mesotrophic outflow from the Everglades. For 2-years from January 2018 to January 2020, we set up 55 dilution and light-dark bottle experiments at five stations to estimate the microzooplankton community grazing rate, instantaneous phytoplankton growth rate, and primary production. Our results suggest that microzooplankton are consuming a higher proportion of the primary production near the Everglades outflow compared to the Florida Keys. We also found that changes in phytoplankton growth rates are disconnected from changes in the microzooplankton grazing rates in the Florida Keys. Overall, the data from the Everglades outflow is what would be expected based on global patterns, but factors other than microzooplankton grazing are more important in controlling phytoplankton biomass in the Florida Keys.

# INTRODUCTION

Phytoplankton form the base of aquatic food webs and produce the majority of autochthonous carbon in aquatic ecosystems. Understanding the fluxes of phytoplankton carbon through marine ecosystems is essential to understand the pathways of primary production. It has been estimated that, on average, 60–75% of marine phytoplankton biomass is consumed daily by microzooplankton (*Calbet & Landry, 2004*), suggesting that a majority of phytoplankton carbon is transferred to higher trophic levels in the pelagic

Corresponding author
Nicole Millette, nmillette@vims.edu

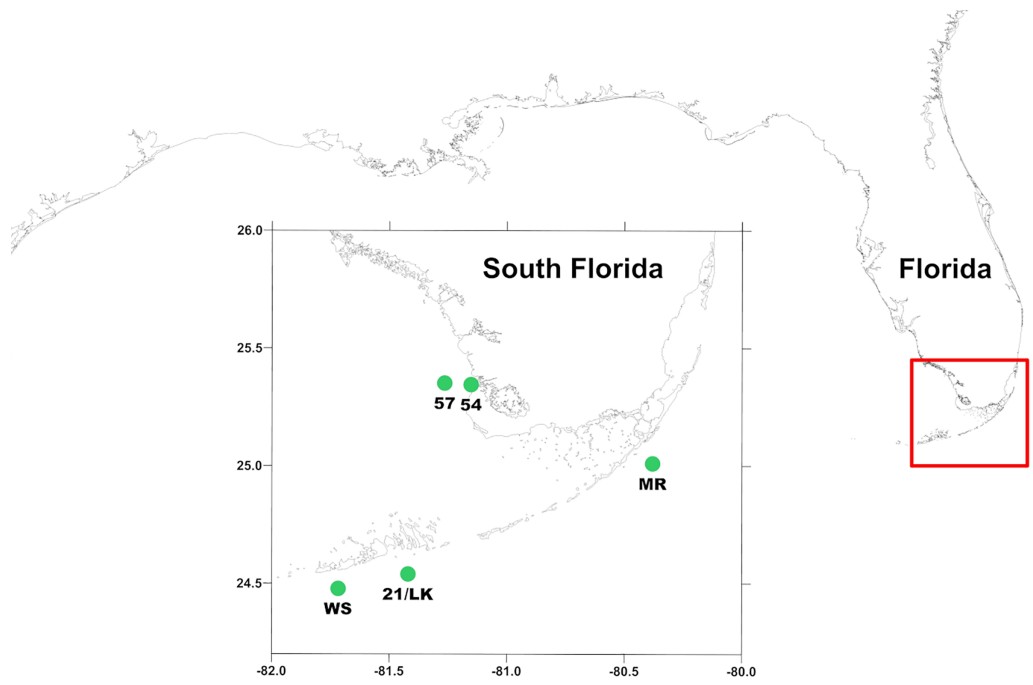

**Figure 1 Location of sampling stations around south Florida.** MR, 21/LK, and WS are located along the coastal region of Florida Keys and 54 and 57 are located in the SWF inner shelf region, where outflow from Shark River enters into the Gulf of Mexico.

food-web. Despite the importance of this predator-prey interaction in mediating phytoplankton population dynamics, there are still large regions of the global ocean where data on microzooplankton grazing activity is missing (*Schmoker, Hernández-León & Calbet, 2013*). This unintended regional focus has limited our understanding of the role of microzooplankton in shaping phytoplankton community structure across a range of marine systems.

One such system is the coastal region around south Florida, USA, from Miami to Ft. Myers (Fig. 1). This region includes the oligotrophic coastal waters of the Florida Keys as well as sediment and nutrient rich river outflow from the Everglades into the Gulf of Mexico. These geographically proximate, but ecologically disparate environments, provide a natural comparison of the role microzooplankton play in mediating the transfer of phytoplankton biomass. This opportunity for regional comparison provides an important view of two understudied subregions. The coastal region of the Florida Keys (FK) is an oligotrophic environment with low nutrient concentrations and phytoplankton biomass (*Boyer & Jones, 2002*). Just north of the FK coastal region is the inner shelf coastal region of southwest Florida (SWF), where freshwater flows out of the Everglades into the Gulf of Mexico from multiple rivers (*Jurado, Hitchcock & Ortner, 2007*). The SWF inner shelf is a mesotrophic environment that has a high input of nutrients, dissolved organic matter, and higher phytoplankton biomass relative to the Florida Keys coastal region (*Boyer & Jones, 2002*; *Jurado, Hitchcock & Ortner, 2007*).

Preliminary analysis of primary production experiments that were conducted on water quality cruises around south Florida between 2014 and 2017 indicated a potential

difference in microzooplankton grazing between FK coastal and SWF inner shelf region. In the FK region, average results between stations were variable ranging from negative to positive rates of net primary production (NPP) and phytoplankton growth, resulting in an opaque picture of the role of microzooplankton play in consuming phytoplankton in this region. Alternatively, in the SWF inner shelf region, average net primary production was positive and the estimated growth rate of phytoplankton in whole water was negative, indicating that while phytoplankton are not limited in growth, microzooplankton consumed a high proportion of the phytoplankton community. However, no direct measurements of microzooplankton grazing rates were measuring in this preliminary analysis time period. To better understand the role of microzooplankton grazers as consumers of phytoplankton biomass in regions of south Florida, we conducted a total of 55 dilution (*Landry & Hassett, 1982*) and primary production experiments at five established water quality stations (three in the FK and two in the SWF inner shelf) around south Florida between January 2018 and January 2020 as part of the ongoing South Florida Ecosystem Restoration Research monitoring cruises.

## MATERIALS AND METHODS

### Sample sites

Measurements were made aboard the *R/V Walton Smith* at five stations (Fig. 1) from January 2018 through January 2020 during South Florida Ecosystem Restoration cruises that occurred every other month. This consisted of three stations along the Florida reef tract—Molasses Reef (MR—25°0′31″N, 80°22′38″W), 21/Looe Key (LK—24°32′08″N, 81°24′74″W), and Western Sambo (WS—24°28′54″N, 81°43′12″W)—and two offshore of the Shark River outflow into the Gulf of Mexico—54 (25°20′42″N, 81°9′43″W) and 57 (25°21′10″N, 80°16′02″W).

### Environmental conditions

Typically, these cruises collect environmental data (temperature, salinity, nutrient and chlorophyll *a* concentrations, etc.) at up to 118 stations. Environmental data were collected using a Sea Bird 911plus Conductivity-Temperature-Depth (CTD) system mounted on a rosette with 12 Teflon-coated 10-L Niskin bottles. Temperature was measured directly from the CTD and salinity was calculated from the conductivity and temperature measurements. The Niskin bottles collected water just below the surface to measure soluble reactive phosphorus (SRP), ammonia, nitrate + nitrite ($NO_x$), and silica concentrations.

Nutrient samples were filtered through a 0.45 μm filter into polystyrene test tubes. The samples were frozen and stored at –20 °C until analysis at NOAA's Atlantic Oceanographic and Meteorological Laboratory (AOML). Nutrients were measured on a SEAL Analytical autoanalyzer using standard gas-segment continuous flow colorimetric methods (*Gordon et al., 1993*). $NO_x$ was measured using a copper-coated cadmium reduction column to reduce nitrate to nitrite. Then, the nitrite was diazotized with sulfanilamide and coupled with N-1-naphthylethylenediamine dihydrochloride to form an azo dye with its absorbance measured at 540 nm (*Zhang, Ortner & Fischer, 1997*). Nitrate concentrations were obtained by subtracting nitrite values, which were measured

separately using the previous procedure without the cadmium reduction, from the $NO_x$ values (*Zhang, Ortner & Fischer, 1997*). Ammonia was measured *via* reaction with alkaline phenol and dichloroisocyanuric acid sodium salt (NaDTT) in the presence of sodium nitroferricyanide as a catalyst at 60 °C to form indophenol blue with its absorbance measured at 640 nm (*Zhang et al., 1997*). To measure SRP, hydrazine was used to reduce 12-molybdophosphoric acid to phosphomolybdenum blue with the absorbance measured at 880 nm (*Zhang , Fischer & Ortner, 2000*). Silicate was measured *via* reaction with molybdate in acidic solution to form β-molybdosilicic acid. The β-molybdosilicic acid was reduced using ascorbic acid to form molybdenum blue with the absorbance measured at 660 nm (*Zhang & Berberian, 1997*).

## Primary production

Primary productivity was estimated at the same five stations mentioned previously *via* changes in dissolved oxygen ($O_2$) concentrations over 24 h using the light-dark bottle method incubated in a continuous-flow water bath with ambient seawater on the deck of the research vessel (*Cullen, 2001*; *Wielgat-Rychert et al., 2017*). Water for the experiments was collected from just below the surface using the Niskin bottles on the CTD, then gently syphoned into 2 L Polycarbonate Nalgene bottles and sealed by rubber stoppers to avoid atmospheric oxygen introduction. Light bottles were covered in one layer of mesh that reduce the ambient light by ~35%, in order to reduce the chance of photoinhibition (*Staehr et al., 2016*). Triplicate samples were collected in 125 mL Erlenmeyer flasks at $T_0$ and $T_{24}$ from the light and dark bottles. Samples were preserved with manganous chloride ($MnCl_2$) immediately followed by the addition of an alkaline sodium hydroxide-sodium iodide solution (NaOH/NaI) and stored in the dark with water level maintained around the glass stopper to minimize oxygen transfer. These samples were then analyzed in the AOML laboratory by Winkler titration using the amperometric technique to estimate the dissolved $O_2$ concentration (*Carpenter, 1995*; *Langdon, 2010*).

Net primary productivity (NPP) was calculated by subtracting the initial $O_2$ concentration from the final $O_2$ concentration in the light bottles. Respiration (R) was calculated by subtracting the initial $O_2$ concentration from the final $O_2$ concentration in the dark bottles. Gross primary productivity (GPP) was calculated by subtracting R from NPP (*Carritt & Carpenter, 1966*; *Duarte, Agustí & Regaudie-de-Gioux, 2011*; *Sanz-Martín et al., 2019*). The change in $O_2$ concentration was converted to carbon, assuming a photosynthetic quotient of 1.28, which is consistent with recent estimates and larger meta-analyses, and a respiratory quotient of 0.89 (*Laws, 1991*; *Wielgat-Rychert et al., 2017*). The change in carbon concentration (μM C) over time was converted to change in biomass per unit volume over time to estimate primary production (gC m$^{-3}$ day$^{-1}$) in the mass of water collected for all experiments.

Duplicate sub-samples of 200 mL were collected to estimate chlorophyll *a* concentrations at $T_0$ and $T_{24}$ from the light bottles. This was used to calculate the change in phytoplankton biomass over the primary production experiments. The 200 mL water samples were filtered onto GF/F filters and stored in liquid nitrogen on the ship and in a −80 °C freezer upon return to the laboratory. Chlorophyll *a* was extracted and measured

within 3 months of collection. The extractions were made in a 60:40 Acetone:Dimethyl Sulfoxide solution and measured on a TD-700 fluorometer following the procedures of *Shoaf & Lium (1976)*. The extractions were then acidified with two drops of 10% HCl and the acidified fluorescence recorded to correct for phaeophytin. The fluorescence values were calibrated using known concentrations of chlorophyll *a* to yield chlorophyll *a* concentration in $\mu$g $L^{-1}$ (*Kelble et al., 2005*). Chlorophyll *a* concentrations in the $T_0$ bottles were used as the estimation of *in situ* concentrations in the surface water.

## Microzooplankton grazing and phytoplankton growth

In addition to all the work normally conducted on these 5-day cruises, experiments were set-up to measure microzooplankton grazing and phytoplankton growth at the same stations where primary production data has been collected since 2014 (Fig. 1). Dilution experiments, as described by *Landry & Hassett (1982)*, were used to measure the community microzooplankton grazing coefficients (Chl *a* Chl $a^{-1}$ $d^{-1}$) on chlorophyll *a* concentrations (phytoplankton biomass) and the instantaneous phytoplankton growth rate (Chl *a* Chl $a^{-1}$ $d^{-1}$). Water for the experiments were collected just below the surface using the Niskin bottles on the CTD. All water was filtered through a 200 $\mu$m mesh to remove all grazers larger than microzooplankton, referred to as whole water hereafter. Filtered water used in the dilution experiments was made using a 0.2 $\mu$m Life Sciences pleated capsule filter. The screened 2 L Nalgene bottle from the primary production experiments (described previously) was used as the 100% whole water treatment. Triplicate 1 L Corning culture flasks were used for each of 20%, 10%, and 5% whole water treatments. The whole and filtered water for each treatment were combined in a 20 L carboy, and then gently transferred to four 1 L flasks (1 initial + 3 final). None of our treatment bottles were amended with inorganic nutrients. The initial flask was used to collect $T_0$ data. The final flasks were placed into 1-layer mesh bags and then incubated on deck in a flow through chamber for 24 h.

Water was collected from all the dilution flasks at the start ($T_0$) and end ($T_{24}$) of our experiments; one sample from the initial treatment flask and one sample from each triplicate $T_{24}$ flasks. Samples were analyzed for chlorophyll *a* as stated above. For each treatment, we used the $T_0$ and $T_{24}$ values of chlorophyll *a* concentrations to calculate the apparent growth rate of phytoplankton biomass using the exponential growth equation (*Landry & Hassett, 1982*). For each experiment, a linear regression was fit to the apparent growth rates for the 10 treatment bottles (100% (1), 20% (3), 10% (3), 5% (3)). The slope of the line (m) was the microzooplankton community grazing coefficient (*g*) and the y-intercept (y) was the phytoplankton instantaneous growth rate ($\mu$) (*Landry & Hassett, 1982*).

We visually assessed the fit of each linear regression and divided the results into four categories; a standard, zero, negative, and non-linear response. A standard response refers to experiments with the expected results, that phytoplankton growth increases as the proportion of whole water decreases (Fig. S1A). The fit of the linear regression for the standard response can vary based on how well all data points fall along the fitted line, but the general pattern is observable. A zero response refers to experiments where minimal

difference between the phytoplankton growth rate at 100% whole water and most dilution treatments were observed (Fig. S1B). A negative response refers to experiments where the phytoplankton growth rate was highest in the 100% whole water treatment and decreased in dilution treatments (Fig. S1C). A non-linear response refers to experiments where the phytoplankton growth rate did not substantially differ from the 100% whole water treatment until the 10% or 5% whole water treatments (Fig. S1D). Regardless of the observed responses, we used a fitted linear regression to obtain our measurements of $\mu$ and $g$ in order maintain consistency across all experiments. We calculated the daily accumulation rate by subtracting $\mu$ from $g$ and the percent of production consumed by dividing $g$ over $\mu$ and multiplying it by 100 (*Calbet & Landry, 2004*; *Anderson & Harvey, 2019*).

### Statistical analysis

The linear regression analysis for all dilution experiments was conducted in Excel using the Analysis ToolPak. Kruskal-Wallis and associated Dunn test analyses to assess differences between stations were conducted in R (version 3.6.3) using the "dunn.test" package (*Dinno, 2017*). These are non-parametric tests because some of our environmental data failed a homogeneity of variance test. We performed a PCA with temperature, salinity, and chlorophyll $a$, $NO_x$, $NH_3$, phosphate, and silicate concentrations collected on each sample data and location to assess the similarity in the environment between each station. The PCA was conducted in R using the built-in function "prcomp". Data from the PCA were extracted and visualized using the "factoextra" package for R, including concentration ellipses with 95% confidence intervals (*Kassambara & Mundt, 2020*).

## RESULTS

### Environmental data

There were numerous differences in environmental factors between the five stations (Table 1), primarily between the Florida Keys (FK—MR, 21/LK, and WS) and southwest Florida inner shelf (SWF—57 and 54). Average chlorophyll $a$ and silicate concentrations were significantly higher at stations 57 and 54 compared to stations MR, 21/LK, and WS. Average salinity at station 54 was significantly lower compared to all stations, while station 57 was significantly lower compared to MR, 21/LK, and WS (Table 1). There were no significant differences in average temperatures, DIN:DIP, and $NO_x$, ammonia and phosphate concentrations. However, station 57 and 54 had noticeability higher concentrations of all nutrients but high variability between sampling dates kept these values from being significantly higher. Stations MR, LK/21, and WS in the coastal region of Florida Keys grouped tightly together and stations 57 and 54 in the SWF inner shelf grouped together (Fig. 2). In all ensuing analyses, these two groupings were separated.

### Primary production experiments

GPP was highest at station 54 followed by station 57, with averages (±SE) of $0.35 \pm 0.08$ gC m$^{-3}$ d$^{-1}$ and $0.17 \pm 0.03$ gC m$^{-3}$ d$^{-1}$ produced, respectively (Fig. 3). GPP rates at both of these stations were significantly higher than 21/LK and WS, and station 54

**Table 1 Average environmental and dilution experiment data.**

| Station | Temp (°C) | Salinity | Chl $a$ (µg L⁻¹) | $NO_x$ (µM) | $NH_3^+$ (µM) | $PO_4^{3-}$ (µM) | $SiO_2$ (µM) | DIN:DIP | $\mu$ (d⁻¹) | $g$ (d⁻¹) | $acc$ (d⁻¹) |
|---|---|---|---|---|---|---|---|---|---|---|---|
| MR | 26.94 ± 0.69[a] | 36.17 ± 0.11[a] | 0.30 ± 0.05[a] | 0.16 ± 0.08[a] | 0.60 ± 0.14[a] | 0.04 ± 0.01[a] | 1.00 ± 0.31[a] | 33.9 ± 12.6[a] | 0.54 ± 0.14[a] | 0.53 ± 0.17[a] | −0.03 ± 0.16[a] |
| 21/LK | 26.52 ± 0.72[a] | 36.13 ± 0.13[a] | 0.51 ± 0.22[a] | 0.18 ± 0.05[a] | 0.84 ± 0.25[a] | 0.04 ± 0.01[a] | 1.26 ± 0.34[a] | 51.7 ± 19.0[a] | 0.85 ± 0.13[a] | 0.21 ± 0.19[a] | 0.02 ± 0.15[a,b] |
| WS | 26.61 ± 0.70[a] | 35.38 ± 0.95[a] | 0.40 ± 0.06[a] | 0.18 ± 0.07[a] | 0.60 ± 0.16[a] | 0.04 ± 0.01[a] | 1.19 ± 0.25[a] | 21.6 ± 8.0[a] | 0.98 ± 0.12[a] | 0.60 ± 0.09[a] | **0.38 ± 0.10[b]** |
| 57 | 25.44 ± 1.55[a] | 35.13 ± 0.35[b] | 2.46 ± 0.73[b] | 0.47 ± 0.35[a] | 1.00 ± 0.21[a] | 0.07 ± 0.02[a] | 19.30 ± 5.37[b] | 35.2 ± 13.2[a] | 0.66 ± 0.17[a] | 0.90 ± 0.21[a] | **−0.24 ± 0.13[a]** |
| 54 | 25.15 ± 1.57[a] | 31.66 ± 0.55[c] | 4.11 ± 0.77[b] | 0.36 ± 0.36[a] | 4.63 ± 2.58[a] | 0.07 ± 0.02[a] | 36.17 ± 9.68[b] | 99.8 ± 49.3[a] | 0.93 ± 0.23[a] | 1.21 ± 0.30[a] | **−0.28 ± 0.11[a]** |

Note:
Average (±SE) temperature, salinity, and chlorophyll $a$, $NO_x$, ammonia, phosphate, silicate concentrations, DIN:DIP, instantaneous phytoplankton growth rate ($\mu$), microzooplankton community grazing coefficient ($g$), and net accumulation rate (acc) at the five sample stations between January 2018 and January 2020. The letters represent stations that are significantly different (paired $t$-test, $P < 0.05$) from each other for each factor. Bold net accumulation rates refers to $\mu$ and $g$ that are statistically different from each other (paired $t$-test, $P < 0.05$), suggesting that the net accumulation rate is either significantly increasing or decreasing.

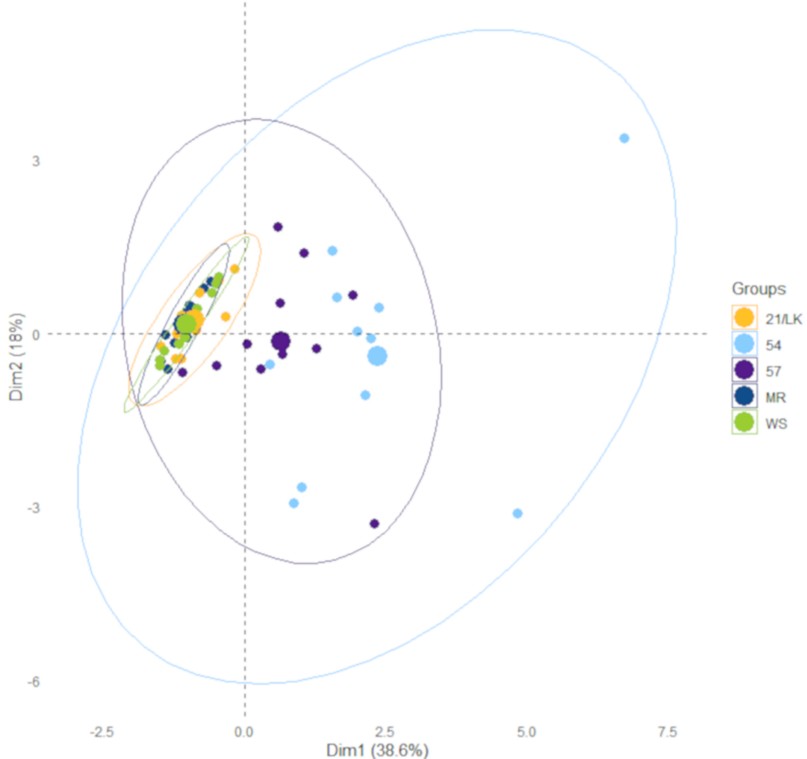

**Figure 2 Results from principle component analysis.** Each data point represents environmental data for each sampling day at the five sample stations between January 2018 and January 2020. The ellipses represent 95% confidence.

was significantly higher than 57. Respiration was highest at stations 57, 54, and MR, with averages (±SE) of −0.16 ± 0.04 gC m⁻³ d⁻¹, −0.14 ± 0.05 gC m⁻³ d⁻¹, and −0.14 ± 0.09 gC m⁻³ d⁻¹ consumed, respectively (Fig. 3). However, rates of respiration were not significantly different between stations. NPP values suggested that, on the whole, the amount of carbon produced and consumed through respiration every day were in balance, with the exception of station 54 (Fig. 3). Average (±SE) NPP at station 54 (0.21 ± 0.07 gC m⁻³ d⁻¹) was significantly higher compared to stations MR and 57, suggesting that more carbon was being produced than consumed through respiration.

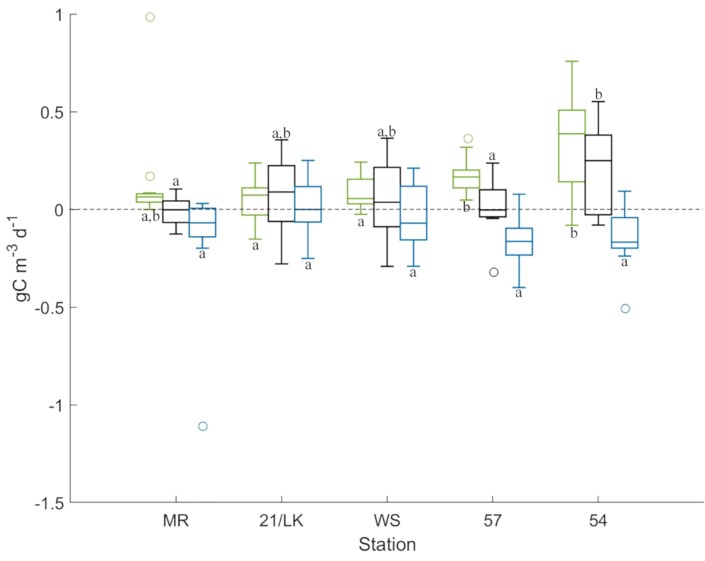

**Figure 3 Results from primary production experiments.** Box plots of GPP (green), NPP (black), and respiration (blue) at each stations from the light-dark primary production experiments. Above the line at 0, carbon is being produced and below 0, carbon is lost. The letters represent stations that are significantly different ($P < 0.05$) from each other for each factor.   

## Microzooplankton grazing

Of the 55 dilution experiments we conducted, three had negative microzooplankton grazing rates and 14 had positive grazing rates that were not significantly different from zero (Tables S1 and S2). In the coastal Florida Keys region, $\mu$ and $g$ appeared to be in balance at stations MR and LK/21 (Figs. 4A and 4C). The average (±SE) accumulation rates at stations MR ($-0.03 \pm 0.16$ d$^{-1}$) and 21/LK ($0.02 \pm 0.15$ d$^{-1}$) were essentially zero, but there was high variability in the accumulation rates between sampling dates (Figs. 4A and 4C). At station WS, the average $\mu$ ($0.98 \pm 0.12$) was significantly higher (paired $t$-test, $P = 0.003$) than the average $g$ ($0.60 \pm 0.09$). Station WS was the only station with an average accumulate rate ($0.38 \pm 0.10$ d$^{-1}$) above zero (Fig. 4C). At all three stations in the coastal region of Florida Keys, there was no linear relationship between $g$ and $\mu$ (simple linear regression, $P$-value > 0.05, not shown).

In the SWF inner shelf region, microzooplankton grazing rates tended to outpace instantaneous phytoplankton growth rates at both stations (Figs. 4B and 4D). The average microzooplankton grazing rates ($0.90 \pm 0.21$) at station 57 was significantly higher (paired $t$-test, $P = 0.04$) than the average instantaneous phytoplankton growth rate ($0.66 \pm 0.17$) and the average microzooplankton grazing rates ($1.21 \pm 0.30$) at station 54 was also significantly higher (paired $t$-test, $P = 0.02$) than the average instantaneous phytoplankton growth rate ($0.93 \pm 0.23$). As a result, the average accumulation rates at stations 57 ($-0.24 \pm 0.13$ d$^{-1}$) and 54 ($-0.28 \pm 0.11$ d$^{-1}$) were below zero (Fig. 4D). At both stations, there was a significantly linear relationship with $g$ and $\mu$, suggesting that $\mu$ and $g$ increase together (simple linear regression, $P$-value < 0.05, not shown).

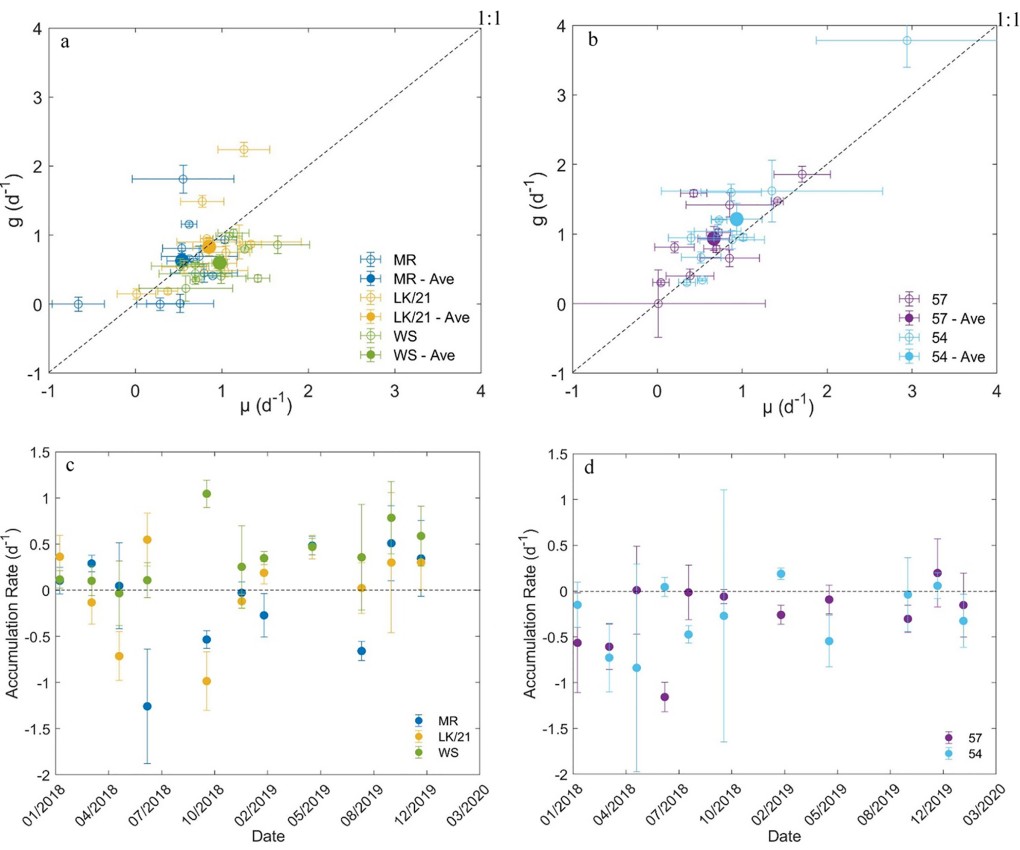

**Figure 4 Results from dilution experiments.** Comparison of the instantaneous phytoplankton growth rate and microzooplankton grazing rate at stations in the (A) coastal region of Florida Keys and (B) SWF inner shelf. The net accumulation rate (±SE) of chlorophyll a concentrations between January 2018 and January 2020 at stations in the (C) coastal region of Florida Keys and (D) SWF inner shelf.

# DISCUSSION

We compared the environmental factors, instantaneous phytoplankton growth rate, microzooplankton community grazing rate, and primary production rate at five stations, representing two distinct environments, over a 2-year period in South Florida. Overall, chlorophyll *a* concentrations (a proxy for phytoplankton biomass) and GPP were substantially higher in the mesotrophic inner southwest shelf (SWF) region, relative to the oligotrophic Florida Keys (FK) region, likely fueled by higher nutrient concentrations in the SWF (Table 1). The two SWF stations, 54 in particular, are located near the outflow of the Shark River from the Everglades in the Gulf of Mexico. The Shark River is known for having some of the highest flow rates in this region, with diatom blooms associated with peak freshwater flow from this river that carries a high nutrient load onto the inner shelf during the wet season (*Jurado, Hitchcock & Ortner, 2007*). Results from the dilution experiments further highlighted differences between the two regions, indicating that the microzooplankton have regional variability in the transfer of carbon up the food-web. The data also indicated high temporal variability in the importance of microzooplankton grazing, particularly in the FK region (Figs. 4A and 4B), but there was no clear driver for

this variability. Therefore, we focus most of our discussion on spatial variability. Our analysis provides the first comparison of community microzooplankton grazing activity between different environments in the South Florida region. More importantly, our analysis highlights additional data we need to collect that can provide important context to our results.

Average microzooplankton community grazing rates ($g$) and instantaneous phytoplankton growth rates ($\mu$) estimated from the dilution experiments were not significantly different between stations. However, the average net accumulation was consistently negative in the SWF region, which differed from the FK region (Figs. 4C and 4D). This means that average $g$ was significantly higher than average $\mu$ at stations 57 and 54. It is uncommon for microzooplankton grazing to be consistently higher than phytoplankton growth (Murrell et al., 2002; Calbet & Landry, 2004). When microzooplankton grazing does exceed phytoplankton growth for an extended period of time it tends to be during periods unfavorable to phytoplankton growth, such as during El Niño in the Pacific Ocean (Verity et al., 1996) or just after peak productivity in the Sargasso Sea (Lessard & Murrell, 1998). While it is unclear why $g$ was higher than $\mu$, the results do suggest that microzooplankton consume a significant proportion of phytoplankton biomass in this region. The linear relationship between $\mu$ and $g$ provides further support that microzooplankton are the primary grazers because it indicates that microzooplankton grazing rates quickly respond to changes in the phytoplankton growth (Calbet & Landry, 2004). On the whole, our analysis suggests that carbon in the phytoplankton pool will primarily be transferred up the food-web through the microzooplankton community in the SWF inner shelf region.

Microzooplankton community grazing in the FK coastal region appears to be consuming less of the phytoplankton biomass. At stations MR and 21/LK average $g$ and $\mu$ are in balance with each other. However, while the average accumulation rates were zero, the accumulation rate was rarely zero for a specific experiment; it tended to fluctuate between positive and negative. We hypothesize that the reason for the higher $\mu$ compared to $g$ at WS and occasionally at MR and 21/LK is that phytoplankton are also consumed by benthic grazers in the FK. Previous research has shown that benthic grazers in our FK study region, such as coral and sponges, are potentially responsible for controlling phytoplankton abundance (Lesser, 2006; Peterson et al., 2006; Wall et al., 2012), putting them in direct competition with microzooplankton. Therefore, we hypothesize during periods of negative accumulation that microzooplankton grazers are the primary consumers of phytoplankton, and during periods of positive accumulation that benthic grazers are the primary consumers of phytoplankton.

Further evidence for this is the lack of coupling between microzooplankton grazing and phytoplankton growth rates at all stations in the FK region. Typically, microzooplankton grazing and phytoplankton growth rates from dilution experiments are linearly related, as at stations 57 and 54, because microzooplankton grazing rates can quickly respond to changes in the phytoplankton growth (Calbet & Landry, 2004). This lack of coupling suggests that microzooplankton are not the only grazers consuming the phytoplankton population in the Florida Keys. This would need to be confirmed by comparing the

abundance of the microzooplankton community between the two regions over time. We would expect in the FK region that microzooplankton abundance would be lower, on average compared to the SWF inner shelf, and that microzooplankton abundance would be lowest when $\mu$ was higher compared to $g$ at any station. This would be the result of benthic grazers taking food away from microzooplankton, thus decreasing their population size and overall community grazing rate measure *via* dilution experiment.

Another possible explanation for the high microzooplankton grazing rates compared to phytoplankton growth rates at stations 54 and 57 could be nutrient limitation caused by the dilution experiments. One assumption of dilution experiments is that phytoplankton growth is not limited by available nutrients (*Landry & Hassett, 1982*). It is common to run parallel dilution experiments, one with nutrients added and one without, to test for and avoid any potential nutrient limitation (*Morison & Menden-Deuer, 2015*; *Anderson & Harvey, 2019*). However, we did not test to see if this assumption was violated in any of our experiments. If phytoplankton at these stations became nutrient limited over the course of the experiments, then we would have underestimated their estimated instantaneous growth rates. If the filtration process broke open cells and released a high concentration of dissolved organic matter into the 0.2 μm filtered water, relieving phytoplankton of their nutrient limitation, then we could have overestimated $\mu$ and $g$. Regardless of whether we underestimated instantaneous phytoplankton growth rates or overestimated microzooplankton grazing rates, we have additional evidence that higher primary production and community respiration is occurring in the SWF inner shelf region compared to the Florida Keys.

While we conducted a large number of dilution and primary production experiments in a region where this type of data has not been collected before, additional data can provide further context to our results. We have already hypothesized that microzooplankton abundance is higher in the SWF inner shelf region compared to the FK coastal region but other factors could be contributing to the spatial and temporal variability in the data. For starters, more information on the phytoplankton and microzooplankton community composition could help describe differences in microzooplankton community grazing rates (*Anderson & Harvey, 2019*). We were unable to collect these data ourselves due to these experiments being conducted as an addition to normal data collection without any additional support. Differences in the phytoplankton and microzooplankton community composition could be another factor driving these differences between regions. In the absence of community composition data, all we can do is speculate how known dominant phytoplankton groups for each region, diatoms in SWF region (*Jurado, Hitchcock & Ortner, 2007*) and cyanobacteria in FK (*Phlips & Badylak, 1996*) could impact microzooplankton grazing. For example, the FK coastal region phytoplankton community tends to be dominated by cyanobacteria because the nutrient concentrations are so low and the system tends to phosphate limited (*Phlips & Badylak, 1996*). Research on microzooplankton grazing rates inside of Florida Bay suggested that as cyanobacteria abundances increased, the impact of microzooplankton grazing decreased (*Goleski et al., 2010*). Therefore, variation in cyanobacteria could be partially responsible for the high amount of variability in microzooplankton grazing rates in this region. However,

microzooplankton have been shown to select against diatoms (*Suzuki et al., 2002*; *Ferreira et al., 2021*). So, it is surprising how consistently high microzooplankton community grazing rates were in the SWF region given that diatoms dominate during certain times of the year (*Jurado, Hitchcock & Ortner, 2007*). The obvious next step is to assess the microzooplankton and phytoplankton communities associated with grazing and growth rates between these regions.

## CONCLUSIONS

The role of microzooplankton in the coastal waters around south Florida has been understudied; our research provided some of the first rates of microzooplankton grazing on phytoplankton for this region. We found evidence that microzooplankton are the dominant phytoplankton grazers in the SWF inner shelf region, as expected, while in the FK microzooplankton and phytoplankton are not as tightly coupled. Given the high phytoplankton biomass and GPP in the SWF inner shelf region, high microzooplankton grazing likely indicates an active pelagic food-web. The low phytoplankton biomass, GPP, and microzooplankton grazing in the FK indicates that the pelagic food-web is less important in this region compared to the SWF. Future research is necessary to understand the pathway that phytoplankton carbon takes in the FK under different conditions. This includes collecting more data on the composition of the phytoplankton community, the abundance of the microzooplankton community, and measuring benthic grazing *vs* microzooplankton on phytoplankton.

## ACKNOWLEDGEMENTS

We would like to thank the crew of the R/V Walton Smith. We would like to thank the lab group's research associate, Charline Quenee as well as the many volunteer student cruise participants. We would also like to thank Dr. Jia-Zhong Zhang and Charles Fischer for conducting the nutrient analyses. This is contribution 4087 of the Virginia Institute of Marine Science, William & Mary.

### Funding

This work was supported by a NOAA/Atlantic Oceanographic and Meteorological Laboratory grant to the Northern Gulf Institute (award number NA160AR4320199). The funders had no role in study design, data collection and analysis, decision to publish, or preparation of the manuscript.

### Grant Disclosures

The following grant information was disclosed by the authors:
Northern Gulf Institute: NA160AR4320199.

### Competing Interests

The authors declare that they have no competing interests.

## Author Contributions

- Nicole Millette conceived and designed the experiments, performed the experiments, analyzed the data, prepared figures and/or tables, authored or reviewed drafts of the paper, and approved the final draft.
- Christopher Kelble conceived and designed the experiments, analyzed the data, prepared figures and/or tables, authored or reviewed drafts of the paper, and approved the final draft.
- Ian Smith performed the experiments, authored or reviewed drafts of the paper, and approved the final draft.
- Kelly Montenero performed the experiments, authored or reviewed drafts of the paper, and approved the final draft.
- Elizabeth Harvey analyzed the data, prepared figures and/or tables, authored or reviewed drafts of the paper, and approved the final draft.

## Data Availability

The raw data are available in the Supplemental File.

## Supplemental Information

Supplemental information for this article can be found online at http://dx.doi.org/10.7717/peerj.13291#supplemental-information.

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

coastal waters by gas segmented continuous flow colorimetric analysis. EPA Method 349.*
Cincinnati: National Exposure Research Laboratory Office of Research and Development, U.S.
Environmental Protection Agency.