# Peer review of "Spatial variability of microzooplankton grazing on phytoplankton in coastal southern Florida, USA"

_PeerJ, doi:10.7717/peerj.13291_

## Round 0.1 · original submission · Major Revisions

Thanks for your interest in publishing in PeerJ.

You should check the comments of the reviewers, carefully and give a response to each concern included in their reviews.

·

Basic reporting

I could find only a few corrections needed for spelling/grammar and basic reporting:

1) check if the label and order of Fig. 5 should be changed to Fig. 4 and vice versa (because the first mention of Fig. 5 is on line 221, and Fig. 4 is later on line 229)

2) line 250: change to "were"

3) line 290: maybe "data" should be treated as a plural, in which case change to "these data"

4) line 297: change to "tends to be"

5) Figure 3: blue, black and red colours can not really be discerned on the reviewing pdf.

6) Table 1 caption: delete "for" in last sentence.

Experimental design

The data here are used mostly for basic description of differences between two regions, and among several stations within those regions, concerning environmental parameters and traits of plankton biology including primary production, respiration and grazing. The introduction describes how the knowledge gap filled using this experimental design is to learn more about plankton variability in an "understudied" region. Although there are many other comprehensive studies on these plankton traits from other regions and the experimental design here is basic, I think it achieves its stated aim and will be useful for informing future related studies in this region.

My only comments are:

line 191: Please provide details of the tests of assumptions of ANOVA, such as Levene's or Cochran's tests, for testing homogeneity of variances.

Table 1: It seems here like only the environmental parameters were statistically compared among regions. For me it was unclear why such comparisons were not extended to the grazing and respiration rates, which in the same way could also be statistically compared among regions. The authors could consider making this addition.

Validity of the findings

no comment

·

Basic reporting

The English used by the authors is professional and the manuscript is generally easy to follow. I provided minor linguistic comments here and there for clarity. The Introduction was simple yet good enough to demonstrate the problem at hand. I would suggest making the objectives a standalone paragraph for clarity but otherwise, it provides context, is well referenced and relevant. I have added some structural points to my comment where needed. The general quality of the figures is not great and makes it hard to distinguish some data points from others, particularly in fig. 4. I suggest removing the back grid in all figures, using black colour for the axis lines and all characters within the figure (instead of light grey as it is now). Labelling is fine unless specifically mentioned. The raw data was not supplied (i.e., Chlorophyll a values versus dilution level) apart from the example figures in Supplementary Information. I do not think that this is an issue as this data is only useful when growth and grazing coefficients are determined. And these are shown unambiguously.

Experimental design

The experimental design was not perfect as, for example, community compositions could have been determined. However, from personal experience, the workload involved in the current experiments and the time required to process and analyse all the samples were already massive as it was. Thus, it would be extremely naïve to ask the authors for more data as their work is commendable. I did highlight a few things on the Methods, though. In addition, I found the Methods to be extremely detailed and believe that the length and detail could be shortened.

Validity of the findings

Even though I did ask for a few tweaks on the author’s analysis that will likely affect the results directly, I think that the fundamental conclusions are not wrong and won’t be affected. Some statistical details are necessary. I think that the conclusions might need a reassessment after addressing the issues that I mentioned in the Discussion.

Additional comments

The Methods are extremely detailed whereas the Discussion could benefit from added information. Aside from a few suggestions in the Methods that will affect the calculations of some figures and the values displayed in the Results, most of my comments focus on the Discussion of the manuscript, which I found a bit insufficient. I will go through each section individually to ensure that each of my comments is easy to understand.

ABSTRACT
The Abstract uses the terminology “Everglades” and “Florida Keys”, whereas the rest of the manuscript compares “SWF” to “FK”. My opinion is that the words are better than the abbreviations as they make the text easier to follow. Also, the words oligotrophic and mesotrophic are used in the Abstract and Introduction but not retained through the rest of the manuscript, namely in the Discussion as this is certainly an important difference between the two regions that must be further highlighted.

INTRODUCTION
Line 57: The abbreviation DOM is never used again; there is no need to abbreviate it.
Line 63: add parenthesis to NPP
Line 72: delete (Fig. 1); you should mention it only in the Methods, I think
Line 74 and 75: I do not see the point of adding results to the Introduction. Delete “Significant (…) locations”.

METHODS
line 81: remove the word “and” before the word “nutrient”
line 78-89: I suggest moving the sentences “Typically (…) since 2014” to different sub-headings of the methods, where environmental data and grazing experiments are described, respectively. The words “at five stations (Fig. 1)” can be moved to the end of the first sentence of this paragraph for clarity.
Line 94, 118, 153: remove the word “from”. Although I must say that it feels odd to repeat this entire sentence three times when one would suffice.
Line 98: re-write as “samples were frozen and stored at -20 (…)”.
Line 101, 106: “Nitrate plus nitrite” should read NOX as per the abbreviation in line 95
Line 105: replace “above” with “previous” – one never knows if it will be located above a given line once the text is formatted to its final form.
Line 107: write the full-name of NaDTT
Line 117: Carpenter (1995) does not mention primary productivity in his paper, nor does he mention the light-dark bottle method. This reference is not suitable to cite here. If something, it should be re-located to line 127 as it specifically addresses the Winkler titration. I could suggest Wielgat-Rychert et al. (2017) as a semi-valid option although there are certainly older works that are more suitable to cite here.
Line 132-134: what is described is the photosynthetic quotient. A more recent estimate (Wielgat-Rychert et al., 2017) suggests that this value is ca. 1.28, not 1 as used by the authors. In addition, I believe that there should be a distinction between respiratory and photosynthetic quotients. Williams and del Giorgio (2005) suggest using an average respiratory quotient of 0.89 after taking into consideration the multiple C sources that can originate CO2 in the cell. Considering that the authors do not know the composition of their communities, I would suggest using these values in their calculations. This implies that GPP must be calculated after converting NPP and R into C-specific values (with the photosynthetic and respiratory quotients, respectively). This modification will affect the results and amends throughout the text and figures will be required.
Lines 150-152: Did the authors use nutrient-amended bottles for their dilution experiments? Not clear and in the methods (although assumed since the original research by Landry and Hassett, 1982 is cited). This is critical for the interpretation of the results (see Calbet and Saiz 2018)
Line 151 and 152: despite being fundamentally correct, displaying grazing and growth coefficients simply as (d-1) is not useful. What the authors are measuring are Chl a-specific coefficients (i.e., Chl a Chl a-1 d-1). I do know that the units nullify but chlorophyll-specific rates are different from cell or C-specific rates, for example (see Ferreira et al., 2021). It is better to write the null units anyway, for clarity. Particularly because GPP is shown in C, not in Chl a.
Line 155: replace “from here on in” with “hereafter”
Line 159 and 160: 20L and four 1L flasks, correct?
Line 161: re-write as “incubated on deck, on a flow (…)”
Line 185: “negative values of g were changed to zero”. Despite having references that support this procedure, I do not agree with it. In fact, a much more recent study by Latasa (2014) specifically recommends not doing so. Negative grazing rates are nothing more than a beneficial influence of the predators over their prey which could be the release of nutrients or the removal of competitors, for example (see Ferreira et al., 2021 and references therein for details). The presence of mixoplankton (see Flynn et al. 2019 for the nomenclature) can also cause these “abnormal” results (see also Ferreira et al., 2021) and, as consequence, these values shouldn’t be neglected nor modified.

RESULTS
Line 201: just add “(Table 1)” after “five stations” and avoid it throughout the rest of the paragraph.
Line 208, 231: remove “amount of”
Line 209: replace “was” with “were”; delete the word “and” before NOX and add a “,” after.
Line 210-211: delete the sentence “However, average NOx concentrations at station 57 and 54 211 had a larger range of variability compared to MR, 21/LK, and WS (Table 1)”. It can be inferred from the table.
Line 211: do not start a sentence with “(Fig. 2)”. Connect the following sentence with the description of the table to improve the readability of the text.
Line 225: add the word “that” before “we ran”.
Line 227 and 228: delete “The three negative (…) not changed (SI Table 1,2)”.

Line 230: this is the first time that accumulation rates are mentioned. The authors use the abbreviation “acc” in Table 1 and in SI but used always the entire words in the manuscript. I suggest changing it to read “accumulation rate” everywhere.
Line 234-235: delete “At all three stations in the coastal region of Florida Keys, there was no linear relationship between g and μ (Fig. 5a).”
Line 243-244: delete “At both station, there was a significantly linear relationship with g and μ, suggesting that μ and g increase together (Fig. 5b).”

DISCUSSION
I believe that the topic of your Discussion that starts on line 285 should be the sentiment that leads the entire Discussion. I’m not saying that the authors should start the Discussion here but the overall sentiment should be directed towards the community analysis. I know that it was impossible to have a detailed community analysis (and I am not criticising it), but it could be a determining factor to explain the differences that are mentioned in the Results. For example, if the FK is dominated by diatoms, microzooplankton grazing rates will certainly decrease as they often display a negative selectivity towards diatoms (e.g., Burkill et al., 1984; Suzuki et al., 2002; Calbet et al., 2011; Ferreira et al., 2021). Nevertheless, you discard this option at the beginning of the Discussion, when diatom blooms are mentioned only in the SWF area.
I also believe that mixotrophy could be part of the reason why some dilution experiments yielded negative or zero grazing rates though I haven’t seen any possible explanation for these events in the discussion (whether related to mixotrophy or not). Are there any studies that show the proportion of mixotrophs in this area? If so, are they dominant? Fig. 2 and 3 haven’t been mentioned at all in the Discussion or Conclusion. If they are earlier figures in the Results, they should be individually discussed before Fig. 4.
Line 250: chlorophyll a is not the same as phytoplankton biomass. It is a proxy for phytoplankton biomass (Landry and Hassett, 1982). And not necessarily a good one due to inter-species differences among other factors (Kruskopf and Flynn, 2006; Ferreira et al., 2021).
Lines 251-252: reference for nutrient concentrations? Is it table 1? There is a clear difference in SiO2 (table 1) but not in NOx, DIN:DIP or in PO4. NH4+ is tendentially higher but the differences are not completely region exclusive (e.g., 21/LK = 54 and 57). Suggest re-writing for clarity.
Line 279: I do not understand the connection with benthic grazers here. You state in the methods that the samples were filtered through a 200µm mesh. There couldn’t be any benthic grazers in your experiments. I do understand and follow the rationale through the rest of the paragraph, but I would be tempted to suggest (without any evidence to support it) that if benthic grazers are such important members of the community, upon getting rid of them, microzooplankton would seize the opportunity and feed more than usual. See the general comment on the Discussion as a community analysis could partly resolve this issue.

CONCLUSION
After applying the suggestions I made to the Methods, Results, and Discussion, I believe that the Conclusion will require a bit of polishing. As I do not know exactly where how will the authors develop the ideas of the manuscript, I am afraid that I cannot make very specific and detailed comments on this section.

Line 310: versus?
Line 310-312: This is a very poetic sentence (e.g., “avenue of mortality”) but is hard to understand. Suggest re-writing it.
* * *
Fig. 1.
I believe that the quality of Fig. 1 can be severely improved. I think that the reader would benefit from having a wide perspective (say of the entire Gulf of México) with a close-up of the sampling area with all 4 sampling stations marked in the close-up perspective. This strengthens the fact that the study was conducted in an under-sampled location. Also (although just an opinion), I would suggest making it look like a drawing instead of a Google Earth snapshot as in Batista et al. (2014) or Gutiérrez-Rodríguez et al. (2015), for example. Colours can still be used if desired.
Fig. 2.
I would suggest keeping only panel a instead of having both. The information is the same and the reader can always go back to Fig. 1 to confirm the region where a given station is located. Also, the method used to determine the statistical ellipses is not mentioned in the methods. If a standard bayesian ellipse was used and its area compared, I would suggest referencing Jackson et al. (2011). This paper dwells on isotopic niches but the statistics should be similar. Regardless of the statistical method employed, this information should be added to the Methods.
Fig. 3.
Must be re-calculated considering respiratory and photosynthetic quotients as I suggested before.
Fig. 4
Is Ave = average? It is not defined in the legend. I would delete panels a and b and make a table instead (see my comment on Tables SI 1 and 2).
Fig. 5
Delete. The information is already shown in figure 4 and nothing new can be inferred from this figure. If necessary, the significant linear fits can be added to figure 4a and 4b. But, as mentioned before, a Table would suffice.
* * *
Table 1.
In the legend, where statistical differences are mentioned, the statistical test and the exact P-value should be referred. The letters µ and g are italicized many times in the manuscript but not here. Revise it. Statistical differences are typically highlighted with lower case letters, not upper case ones, if I am not mistaken.
* * *
Fig. SI 1-4
Examples are ok to show but are not very useful because the Methods have a very thorough explanation of what is a standard dilution result and what isn’t. If the idea is simply to show the different slopes as an example, a single figure with 4 panels is enough. Each panel could depict one scenario instead of having four 4-5 panelled figures.
Fig. SI 4. These are typical responses to food saturation. In these cases, the growth and grazing coefficients should be determined as follows: phytoplankton growth rates (µ, Chl a Chl a−1 h−1) are determined from the interception of the linear regression with the most diluted treatments. The grazing coefficients (g) are then calculated as g = µ−K where K (Chl a Chl a−1 h−1) is the apparent growth rates obtained in the undiluted bottles according to Gallegos (1989) and Dolan et al. (2000). This information should also be added to the Methods under the heading “Microzooplankton grazing and phytoplankton growth”.
* * *
Tables SI 1 and 2
I believe that this information should be included in the main manuscript (negative values included and specified – Latasa 2014) as a single table. Ultimately, one can extract this data from Fig. 4a and 4b but it is easier if a table is provided instead.

Good luck,
Guilherme D. Ferreira

REFERENCES CITED
Batista, M.I., Henriques, S., Pais, M.P., Cabral, H.N., 2014. Assessment of cumulative human pressures on a coastal area: Integrating information for MPA planning and management. Ocean Coast Manag 102, Part A, 248-257.
Burkill, P.H., Mantoura, R.F.C., Llewellyn, C.A., Owens, N.J.P., 1987. Microzooplankton grazing and selectivity of phytoplankton in coastal waters. Mar. Biol. 93, 581-590.
Calbet, A., Saiz, E., 2018. How much is enough for nutrients in microzooplankton dilution grazing experiments? J. Plankton Res. 40, 109-117.
Calbet, A., Saiz, E., Almeda, R., Movilla, J.I., Alcaraz, M., 2011. Low microzooplankton grazing rates in the Arctic Ocean during a Phaeocystis pouchetii bloom (Summer 2007): fact or artifact of the dilution technique? J. Plankton Res. 33, 687-701.
Dolan, J.R., Gallegos, C.L., Moigis, A., 2000. Dilution effects on microzooplankton in dilution grazing experiments. Mar. Ecol. Prog. Ser. 200, 127-139.
Ferreira, G.D., Romano, F., Medić, N., Pitta, P., Hansen, P.J., Flynn, K.J., Mitra, A., Calbet, A., 2021. Mixoplankton interferences in dilution grazing experiments. Sci. Rep. 11, 23849.
Flynn, K.J., Mitra, A., Anestis, K., Anschütz, A.A., Calbet, A., Ferreira, G.D., Gypens, N., Hansen, P.J., John, U., Martin, J.L., Mansour, J.S., Maselli, M., Medić, N., Norlin, A., Not, F., Pitta, P., Romano, F., Saiz, E., Schneider, L.K., Stolte, W., Traboni, C., 2019. Mixotrophic protists and a new paradigm for marine ecology: where does plankton research go now? J. Plankton Res. 41, 375-391.
Gallegos, C.L., 1989. Microzooplankton grazing on phytoplankton in the Rhode River, Maryland: Nonlinear feeding kinetics. Mar. Ecol. Prog. Ser. 57, 23-33.
Gutiérrez-Rodríguez, A., Selph, K.E., Landry, M.R., 2015. Phytoplankton growth and microzooplankton grazing dynamics across vertical environmental gradients determined by transplant in situ dilution experiments. J. Plankton Res. 38, 271-289.
Jackson, A.L., Inger, R., Parnell, A.C., Bearhop, S., 2011. Comparing isotopic niche widths among and within communities: SIBER – Stable Isotope Bayesian Ellipses in R. J. Anim. Ecol. 80, 595-602.
Kruskopf, M., Flynn, K.J., 2006. Chlorophyll content and fluorescence responses cannot be used to gauge reliably phytoplankton biomass, nutrient status or growth rate. New Phytol. 169, 525-536.
Landry, M.R., Hassett, R.P.l., 1982. Estimating the grazing impact of marine micro-zooplankton. Mar. Biol. 67, 283-288.
Latasa, M., 2014. Comment: A potential bias in the databases of phytoplankton growth and microzooplankton grazing rates because of the improper formulation of the null hypothesis in dilution experiments. Limnol. Oceanogr. 59, 1092-1094.
Suzuki, K., Tsuda, A., Kiyosawa, H., Takeda, S., Nishioka, J., Saino, T., Takahashi, M., Wong, C.S., 2002. Grazing impact of microzooplankton on a diatom bloom in a mesocosm as estimated by pigment-specific dilution technique. J. Exp. Mar. Biol. Ecol. 271, 99-120.
Wielgat-Rychert, M., Rychert, K., Witek, Z., Zalewski, M., 2017. Calculation of the photosynthetic quotient (PQ) in the Gulf of Gdańsk (Southern Baltic). Balt. Coast. Zone 21, 51-60.
Williams, P.J.I.B., del Giorgio, P.A., 2005. Respiration in aquatic ecosystems: history and background, in: del Giorgio, P.A., Williams, P.J.I.B. (Eds.), Respiration in Aquatic Ecosystems. Oxford University Press, Oxford University Press Inc., New York, pp. 1-17.

---

## Round 0.2 · Minor Revisions

After checking the comments of reviewers, a decision has been made, only minor changes (typos should be corrected) and clarifications. I am awaiting your manuscript with the changes.

·

Basic reporting

The only further comments I have are about a small number of minor changes needed for the basic reporting:

line 51: fix typo here

line 134: is this meant to be "meta-analyses"?

line 201: change to "Data from the PCA were extracted"

line 227: change "Fig. 5" to "Fig. 3". Also please capitalise the word directly after this.

line 339: change to "dominant"

line 346: change "about" to "amount". Also, on line 348, add "is" between "it" and "surprising"

In the figures section, the captions in the page immediately before some of the figures is different to the caption added on the same page of each figure (I guess these should be the same). When this manuscript is further processed please make sure the correct/updated version of each caption is used.

Experimental design

no comment

Validity of the findings

no comment

·

Basic reporting

I'm glad to see that the authors have decided to include most of my suggestions in the manuscript. Not because I was the one making them but because I truly believe that they bring the manuscript a notch above. I am particularly happy to see the massive changes in the figures, in the calculation of photosynthetic and respiration data, and in the Supplementary Information. I also believe that the Discussion is now ready to put the author's novel results in a higher perspective and I like the flow of the text and the arguments presented. On the rare occasions where the authors decided not to include my comments, they have provided a sound rebuttal and I am happy with their responses and the references that support them.

In my opinion, the manuscript has been severely improved after the revision process and I am happy to have been a part of it. I honestly hope that the authors agree with me. Therefore, I recommend this manuscript to be accepted for publication in PeerJ as is.

Experimental design

No comment.

Validity of the findings

No comment.

Additional comments

No comment.

---

## Round 0.3 · accepted · Accept

Thanks for submitting the revised version of the ms with the suggested changes.

Best regards,